# Preparation and Physical Properties of High-Belite Sulphoaluminate Cement-Based Foam Concrete Using an Orthogonal Test

**DOI:** 10.3390/ma12060984

**Published:** 2019-03-25

**Authors:** Chao Liu, Jianlin Luo, Qiuyi Li, Song Gao, Zuquan Jin, Shaochun Li, Peng Zhang, Shuaichao Chen

**Affiliations:** 1School of Civil Engineering, Qingdao University of Technology, Qingdao 266033, China; alexlc@163.com (C.L.); gaosong727@126.com (S.G.); jinzuquan@126.com (Z.J.); 15063016520@126.com (S.L.); zhp0221@163.com (P.Z.); 17853245963@163.com (S.C.); 2Collaborative Innovation Center of Engineering Construction and Safety in Shandong Blue Economic Zone, Qingdao University of Technology, Qingdao 266033, China; 3Center for Infrastructure Engineering, School of Computing, Engineering and Mathematics, Western Sydney University, Sydney, NSW 2751, Australia; 4School of Architecture Engineering, Qingdao Agricultural University, Qingdao 266109, China

**Keywords:** foam concrete, high-belite sulphoaluminate cement, dilution ratio, dry density, compressive strength, thermal insulation, microstructure

## Abstract

Prefabricated building development increasingly requires foam concrete (FC) insulation panels with low dry density (*ρ*_d_), low thermal conductivity coefficient (*k*_c_), and a certain compressive strength (*f*_cu_). Here, the foam properties of a composite foaming agent with different dilution ratios were studied first, high-belite sulphoaluminate cement (HBSC)-based FCs (HBFCs) with 16 groups of orthogonal mix proportions were subsequently fabricated by a pre-foaming method, and physical properties (*ρ*_d_, *f*_cu_, and *k*_c_) of the cured HBFC were characterized in tandem with microstructures. The optimum mix ratios for *ρ*_d_, *f*_cu_, and *k*_c_ properties were obtained by the range analysis and variance analysis, and the final optimization verification and economic cost of HBFC was also carried out. Orthogonal results show that foam produced by the foaming agent at a dilution ratio of 1:30 can meet the requirements of foam properties for HBFC, with the 1 h bleeding volume, 1 h settling distance, foamability, and foam density being 65.1 ± 3.5 mL, 8.0 ± 0.4 mm, 27.9 ± 0.9 times, and 45.0 ± 1.4 kg/m^3^, respectively. The increase of fly ash (FA) and foam dosage can effectively reduce the *k*_c_ of the cured HBFC, but also leads to the decrease of *f*_cu_ due to the increase in mean pore size and the connected pore amount, and the decline of pore uniformity and pore wall strength. When the dosage of FA, water, foam, and the naphthalene-based superplasticizer of the binder is 20 wt%, 0.50, 16.5 wt%, and 0.6 wt%, the cured HBFC with *ρ*_d_ of 293.5 ± 4.9 kg/m^3^, *f*_cu_ of 0.58 ± 0.02 MPa and *k*_c_ of 0.09234 ± 0.00142 W/m·k is achieved. In addition, the cost of HBFC is only 39.5 $/m^3^, which is 5.2 $ lower than that of ordinary Portland cement (OPC)-based FC. If the surface of the optimized HBFC is further treated with water repellent, it will completely meet the requirements for a prefabricated ultra-light insulation panel.

## 1. Introduction

Since ‘sustainability’ was widely adopted as a key criterion for the assessment of construction materials and buildings [1,2,3], researchers around the world have realized the growing demand for lightweight, economical, easy-to-use, and environmentally sustainable building materials in the future [4,5,6,7,8]. Foam concrete (FC) products with low self-weight, high specific strength, and excellent thermal insulation performance become very attractive in the application of prefabricated building panels [7,9]. FC products can effectively reduce dead loads on the structure and foundation, contribute to energy conservation, and lower the labor cost during construction [10,11]. Nowadays, FC has been commonly used in construction applications in different countries such as Germany, UK, Philippines, Turkey, Thailand, and China [12,13].

FC is generally produced with cement, filler, water, and a liquid chemical under controlled conditions, and the liquid chemical can be diluted by water and aerated to foaming [14,15,16,17,18,19,20,21,22]. Although the dry density (*ρ*_d_) and thermal conductivity coefficient (*k*_c_) of FC is lower than that of ordinary concrete, its insufficiency of high porosity, multiple connected pores, and low strength limits its widespread development. Recently, a lot of research has been done to further improve the performance of FC [23,24,25,26,27,28,29,30,31]. Kearsley and Wainwright found that replacing cement with a high content of fly ash (FA) could enhance the late strength of FC [26]. Khan et al. found that polypropylene fiber (PP) could increase the flexural strength (*f*_b_) and tensile strength (*f*_t_) of FC, but it had no effect on compressive strength (*f*_cu_), while basalt fiber could greatly increase *f*_cu_, *f*_b_, and *f*_t_ of FC, which was better than PP [27]. Luo et al. reported that adding multi-walled carbon nanotubes (CNTs) to FC could improve pore structure and reduce average pore size [28]. Yakovlev et al. found that adding CNTs to FC could significantly increase its *f*_cu_ and effectively improve its heat insulation [29]. Szelag introduced CNTs dispersion and sodium dodecyl sulfate (SDS) into the cement slurry and found that a large amount of foam was generated after fast stirring due to the foaming characteristics of SDS, which greatly reduced the density, and concluded that CNTs reinforced cement paste with SDS was expected to be used in the production of lightweight concrete [30]. Sun et al. explored the effects of a synthetic surfactant, plant surfactant, and animal glue/blood-based surfactant on the properties of FC and found that FC prepared by the synthetic surfactant had a higher *f*_cu_ and smaller shrinkage [31].

From the findings of the above studies, the performance of FC is still does not meet the requirements for a prefabricated insulation wall. In addition, due to the slow setting feature of ordinary Portland cement (OPC), it is difficult to match the defoaming time of the foam [3,17], frequently resulting in the collapse of the FC. Moreover, the early-stage strength of FC prepared with OPC increases slowly, which is not conducive to its application in the prefabricated industry and affects the engineering efficiency [32]. Unlike sulphoaluminate cement (SAC), which uses a lot of expensive bauxite, high-belite sulphoaluminate cement (HBSC) can be mostly calcined by construction and industrial waste, which is consistent with the concept of sustainability [33,34,35,36,37,38,39]. HBSC has many advantages, such as fast setting, fast hardening, early strength, high strength, small expansion, low dry shrinkage, anti-freezing, anti-permeability, anti-corrosion, and so on [36]. Indeed, nowadays HBSC is widely used, allowing that HBSC-based FC (HBFC) has better thermal insulation performance than ordinary concrete and can be used for insulation panels to realize the energy-saving efficiency of buildings [20].

HBFC can be used to control the stability of bubbles in FC, and high strengths can be attained in the initial stages of curing. Therefore, the preparation of FC with HBSC is conducive to the optimization of properties.

There are two techniques that could be used in the FC process; the pre-foaming method and mix-foaming method [40,41,42,43]. The pre-foaming method is to make the foam before mixing it with slurry and, after mixing evenly, the FC slurry is poured into the mold to form an FC product. The mix-foaming method is to prepare a slurry containing a foaming agent first, then pre-cast the slurry and complete the foaming during incubation [36]. The fresh FC prepared by the pre-foaming method has good fluidity and can be pumped to a long distance, which meets the prefabricated process, while the mix-foaming method is generally not used for pumping and just for on-site pouring.

Orthogonal experiments can be used to effectively explore the relative importance of various factors on the performance of HBSC-based FC (HBFC). The optimum level of different factors can be determined by orthogonal arrays. The use of an orthogonal design can significantly reduce the cost and time of the experiment [44,45]. The mixture ratio of HBFC with stable and excellent properties can be selected more accurately by using an analysis of means (ANOM) and variances (ANOVA) [14,28,29].

Here, in order to seek the optimum mixture ratio of HBFCs meeting the performance requirements of modern prefabricated lightweight panels, mix proportions of 16 groups HBFCs were designed with an orthogonal experiment, and the HBFCs were fabricated by the pre-foaming method. The corresponding *ρ*_d_, *f*_cu_, and *k*_c_ were characterized and analyzed by ANOM and ANOVA, the optimization verification of the mix ratio and economic cost of HBFC were also evaluated.

## 2. Materials and Methods

### 2.1. Raw Materials

The HBSC, acquired from Polar Bear Building Material Co., Ltd. (Tangshan, China), was produced with solid waste. Grade-II fly ash (FA) was obtained from the Shandong Huadian power plant (Qingdao, China). The chemical composition and mineral composition of the HBSC clinker and the chemical composition of FA were characterized by X-ray powder diffraction (XRD, D8 Advance type, Bruker Corp., Leipzig, Germany) and an X-ray fluorescence spectrometer (XRF, XRF-1800 type, Shimadzu Corp., Kyoto, Japan), listed in Table 1 and Table 2. The composite foaming agent was bought from Guangzhou Haofeng chemical company (Guangzhou, China). The naphthalene-based superplasticizer (NSP) was bought from Shanghai Chenqi chemical company (Shanghai, China). The tap water was used. 

### 2.2. Mix Proportion and Orthogonal Experimental Design

The dosage to binder of FA, water, foam, and the naphthalene-based superplasticizer (*W*_FA_, *W/B*, *w*_FOAM_, and *w*_NSP_) are the main parameters separately or jointly affecting the physical performances (*ρ*_d_, *f*_cu_, and *k*_c_) of HBFC. The mix proportions of FCs were designed using an orthogonal experiment with four parameters *w*_FA_, *W/B*, *w*_FOAM_, and *w*_NSP_ as the main factors. Sixteen different mixtures of HBFCs were prepared by varying (1) FA replacement for HBSC (*w*_FA_) from 0 to 20 wt% with a 5 wt% gradient, (2) water to binder ratio (*W*/*B*) from 0.40 to 0.55 with a 0.05 gradient, (3) foam fraction of binder (*w*_FOAM_) from 13.5 to 16.5 wt% with a 1 wt% gradient, (4) NSP fraction of binder (*w*_NSP_) from 0 to 1.0 wt% with a 0.2 wt% gradient.

The *k*_c_ of FA is lower than cement, which can improve the thermal insulation of FC, and its pozzolanic effect can improve the long-term strength of FC. The superplasticizer can improve the workability of the fresh FC slurry, reduce the *W*/*B*, and eventually improve the *f*_cu_ of FC. The amount of foam affects the *ρ*_d_ and *f*_cu_ of the FC—the greater the *w*_FOAM_, the greater the pore volume and the lower the *k*_c_ of the FC. Four levels were taken for each factor, as listed in Table 3 and Table 4 [28]. The *ρ*_d_, *f*_cu_, and *k*_c_ were used as evaluation indicators and the optimal mix ratio was finally achieved by the ANOM and ANOVA methods.

### 2.3. Preparation Procedures of HBFC

First, the HBSC, FA, and NSP were weighed according to the designed mix ratio and dry-mixed in a rotating mortar mixer (JJ-5 type, Wuxi, China) for 1 min. Then, the weighed water was added into the mixer to prepare the slurry. Meanwhile, the appropriate weight of foam was generated by a foam generator (BL168-8 type, Hefei Baile Energy Equipment Company, Hefei, China) and immediately added to the slurry mixture. The foam was mixed with the slurry and stirred for some time until there was no physical indication of the foam on the surface and all the foam was evenly distributed and incorporated into the mixture [36].

Fresh HBFC slurry was poured into six cubic molds with sizes of 100 × 100 × 100 mm^3^ and three prism molds with sizes of 300 mm × 300 mm × 60 mm. Unlike a normal concrete casting process, the HBFC slurry should not be subjected to any form of compaction or vibration. The specimens were cast in molds covered with plastic film to prevent moisture loss. Once demolded after 24 h, the specimens were kept in a standard curing room (temperature 22 ± 2 °C, relative humidity ≥95%) up to the day of testing. The schematic fabricating process of HBFC is shown in Figure 1.

## 3. Property Characterizations

### 3.1. Properties of Foam

Taking into consideration the cost of the foaming agent and its foaming performance, it was necessary to select an appropriate dilution ratio to configure the foaming agent solution, which was used to produce HBFC [35]. In this study, the above-mentioned composite foaming agent was used to conduct four tests with varied dilution ratios of 1:10, 1:20, 1:30, and 1:40, respectively, and its foam properties were tested using a foam performance measuring instrument (YJ-1 type, foam tester for FC, Tianjian Instrument Co., Ltd., Cangzhou, China), shown in Figure 2.

The foam properties were characterized in accordance with criterion JG/T 266-2011 (Beijing, China). The volume of the ⑤capacity cylinder is *V*_F_, and its mass was recorded as *m*_a_ after being measured. The foam was directly sprayed into the ⑤capacity cylinder through the aspiration pipe of the foam generator. The excess foam on the top of the ⑤capacity cylinder was scrapped and the total mass of the foam and the ⑤capacity cylinder *m*_b_ was weighed. The foam density (*ρ**_F_*) can be calculated by Equation (1) and the foamability can be calculated by Equation (2):(1)$$\rho_{F}=\frac{m_{b}-m_{a}}{V_{F}}$$
(2)$$M=\frac{V_{F}}{(m_{b}-m_{a})/\rho }$$
where *M* is the foamability of the foam, *V*_F_ is the volume of the foam, *m*_b_ is the total mass of the foam and the capacity cylinder, *m**_a_* is the mass of the foam, *ρ**_F_* is the foam density, and *ρ* is the density of the foaming agent solution.

After measuring the *ρ*_F_ and *M*, the capacity cylinder filled with foam was placed on top of the ①steel tray. After 1 h, the 1 h settling distance of foam in the ⑤capacity cylinder and 1 h bleeding volume of foam in the ⑦bleeding capacity bottle were recorded as *S*_v_ and *B*_v_, respectively. The foam properties of each foam agent dilution ratio were measured 3 times.

### 3.2. Dry Density

The cubic specimens were weighed after they were dried at 55 ± 5 °C to constant weights within an oven drier (BG2-140 type, vacuum oven, Shanghai Boyi Equipment Factory, Shanghai, China), then the *ρ*_d_ of the specimen was calculated from the weight to volume ratio and three duplicates for each group were conducted.

### 3.3. Compressive Strength

Since the strengths of HBFC were relatively low, these cubic specimens were crushed using a microelectronic control testing machine (YAW-3000 type, Jinan Nusino Industrial Testing System Co., Ltd., Jinan, China) with 1.0 mm/min cross head speed and nearest 0.01 N precision [11], and three duplicates for each group were conducted. The *f*_cu_ was acquired from Equation (3):*f*_cu_ = *F*/*A*_c_(3)
where *f*_cu_ is the compressive strength of the HBFC specimen at 7 d age, *F* is the maximum failure load, and *A*_c_ is the initial pressure area of the test specimen.

### 3.4. Thermal Conductivity

The *k*_c_ of the material is the heat transferred per unit of area and per unit of time when the temperature difference between the two sides is 1°C under stable heat transfer conditions. Here, the prism HBFC panels (300 mm × 300 mm × 60 mm) were dried at 55 ± 5 °C to a constant weight in order to eliminate the influence of moisture on *k*_c_ then polished on both sides to achieve good contacts between specimens and hot (or cold) plates. An automatic double plane thermal conductivity tester (SSX-DR300 type, thermal conductivity measuring instrument, Beijing Sansixing Measurement and Control Technology Co., Ltd., Beijing, China) was employed to measure *k*_c_ of HBFC. The HBFC specimen was placed between two plates, the temperatures of the hot and cold plates were set as 40 °C and 20 °C, and the heat flow rate was measured when the equilibrium condition was reached after about 3 h [36]. The temperature difference between hot and cold plates was controlled by a control unit [37]. Three duplicates for each group were examined, and the *k*_c_ of the HBFC can be calculated by Equation (4):(4)$$k_{c}=\frac{\varphi \times d}{S\times \Delta T}$$
where, *k*_c_ is the thermal conductivity (W/m·k), *φ* is heat flow rate (J/s), *d* is average thickness of the specimen (m), *S* is average area of the specimen (m^2^), and *ΔT* is temperature difference between the hot and cold plates (°C).

### 3.5. Macroscopic and Microscopic Observation

The pore sizes, distributions, and pore structure in the macroscopic and microscopic scale of HBFC were observed by a digital camera (EOS 800D type, Canon (China), Beijing, China) and field emission scanning microscope (FEG-SEM, JSM 7500f type, Jeol, Tokyo, Japan), respectively.

## 4. Results

### 4.1. Foam Properties

The mean results and deviations of the foam properties of composite foam agents under different dilution ratios are revealed in Table 5. The basic requirements for the performance index of the foam agent can refer to criterion JG/T 266-2011 (Beijing, China), also shown in Table 5.

As shown in Table 5 and Figure 3, the 1 h bleeding volume and the 1 h settling distance were lowest (40.6 ± 2.1 mL and 4.7 ± 0.3 mm) and highest (89.3 ± 4.0 mL and 11.5 ± 0.6 mm) when the dilution ratio was 1:10 and 1:40, respectively. With the increase of dilution ratio, the 1 h bleeding volume and 1 h settling distance of the foam steadily increased, as shown in Figure 3. In fact, as the dilution ratio increased, the water content of the foam liquid membrane also increased and the tenacity of the liquid membrane decreased, cracking and bleeding were thus more likely to occur—resulting in defoaming [43].

Foamability was highest (35.1 ± 1.2 times) and lowest (23.7 ± 1.4 times)—and the foam density was the lowest (29.5 ± 0.9 kg/m^3^) and highest (49.1 ± 1.4 kg/m^3^)—when the dilution ratio was 1:10 and 1:40, respectively, as demonstrated in Table 5 and Figure 4. With the increase of dilution ratio, foamability gradually decreased and the foam density gradually increased, as shown in Figure 4. Indeed, as the dilution ratio increased, the concentration of the foam solution decreased, resulting in an increase in the surface tension of the liquid and a decrease in the foamability [44,45]. The water content in the foam liquid membrane increased and its density increased accordingly.

Thus, considering the cost and performance requirements of composite foam agents, the dilution ratio was selected at 1:30 to produce the foam for HBFC.

### 4.2. Physical Properties and Orthogonal Range Analysis

The mean results and deviations of the physical properties of 16 groups of HBFCs are revealed in Table 6. An ANOM of the orthogonal test and mix proportion optimization of HBFC are shown accordingly in Table 7.

From the physical properties and the corresponding ANOM shown in Table 6 and Table 7, the optimal mix proportion can be obtained with the primary and secondary factors *w*_FOAM_ and *w*_NSP_, respectively. The optimal ratio to reach the lowest *ρ*_d_ is *A*_2_*B*_1_*C*_4_*D*_1_ with *w*_FA_, *W*/*B*, *w*_FOAM_, and *w*_NSP_ as 10 wt%, 0.4, 16.5 wt%, and 0 wt%, respectively.

After ANOM calculations, the order of the factors affecting *f*_cu_ is obtained, which can be ranked in order of importance as *D* > *A* > *C* > *B*. Finally, optimal mix proportion can be obtained at *A*_1_*B*_2_*C*_2_*D*_2_ with the value of four factors as 0 wt%, 0.45, 14.5 wt%, and 0.6 wt%, respectively.

From the ANOM, the optimal level of *k*_c_ can be obtained, optimal mix proportion is *A*_4_*B*_3_*C*_4_*D*_1_ with *w*_FA_, *W*/*B*, *w*_FOAM_, and *w*_NSP_ as 20 wt%, 0.5, 16.5 wt%, and 0 wt%, respectively, which can be ranked in order of importance as *A* > *C* > *D* > *B*.

Specific analysis on each indicator is further demonstrated as follows.

### 4.3. Effect of Various Factors on Dry Density

The relationship between *ρ*_d_ and four factors is shown in Figure 5. Even though the mix proportion of HBFC is designed based on the predetermined *ρ*_d_ value of 350 kg/m^3^, the minimum density can be obtained within this narrow density range. The HBFC has the lowest *ρ*_d_ when *w*_FA_ is 10 wt% because the setting time of the binder just matches the foam stabilization time, which minimizes defoaming [40]. It can be seen from Figure 5 that *ρ*_d_ first increased and then decreased with the increase of *W*/*B*. With the increase of water, the bubble of the foam gradually tends to be saturated, and much of the foam with thin bubble walls will burst due to the unbearable pressure, resulting in a decrease in the amount of complete foam and an increase in *ρ*_d_. When the water continues to increase however, the burst foam does not continue to increase as the bubbles have already saturated. Water’s participation in the hydration is certain, the extra water will only serve as free water to prop up part of the volume which can be evaporated after drying, thus *ρ*_d_ will be reduced. The *ρ*_d_ of HBFC also first increased and then decreased with *w*_FOAM_. Obviously, the foam made up most of the volume of HBFC, *w*_FOAM_ was found to be the most influential factor on *ρ*_d_, as revealed from the ANOM in Table 7.

This phenomenon can be explained by the SEM images of microscopic pore structure in Figure 6. When the *w*_FOAM_ was 14.5 wt%, the average pore size was smallest within 180 μm and the size distribution was uniform. When the *w*_FOAM_ was 13.5 wt% and 15.5 wt%, the average pore size was 220 μm and 200 μm, respectively. When the *w*_FOAM_ was 16.5 wt%, relatively large pores appeared within the largest one of 450 μm and pore sizes differed greatly. The average pore diameter of HBFC first increased and then decreased with the increase of *w*_FOAM_, which meant that the pore volume in HBFC first increased and then decreased, resulting in the fluctuating trend of *ρ*_d_ with *w*_FOAM_ [37]. With the increase of *w*_NSP_, the *ρ*_d_ of HBFC shows a similar fluctuating trend as with *w*_FOAM_. As reported, appropriate *w*_NSP_ can maintain good stability of the foam by improving slurry fluidity, but it will reduce the plastic viscosity of the slurry, resulting in broken air bubbles [37]. When *w*_NSP_ was 0.6 wt%, only the plastic viscosity of the slurry was reduced, but the fluidity was not effectively improved, which caused a large number of broken bubbles and the *ρ*_d_ of HBFC eventually increased. However, the fluidity of the slurry improved and the slurry hardness decreased as *w*_NSP_ continued to increase, resulting in a decrease in the amount of broken bubbles and a decrease in *ρ*_d_.

### 4.4. Effect of Various Factors on Compressive Strength

As shown in Figure 7, the *f*_cu_ decreased with the increase of *w*_FA_. The phenomenon is similar to that of ordinary concrete. The early strength of HBFC is mainly controlled by cement and the 7d strength of HBSC can reach about 90% of its final strength due to its fast hardening and early high strength properties [11].

Thus, the higher the content of HBSC in HBFC, the higher the *f*_cu_ will be. This rule can also be explained by the microstructure of HBFC—when the binder is only HBSC, the pores of FC are almost closed, there are few connected pores, and the structure of the pore walls is very tight and ductile. With the *w*_FA_ increase, the number of damaged connected pores tended to increase and the structure of the pore wall gradually became loose, as shown in Figure 8. Thus, *f*_cu_ decreased as *w*_FA_ increased.

### 4.5. Effect of Various Factors on Thermal Conductivity

Figure 9 presents the effect of four factors and levels on the *k*_c_ of HBFC. Among the four factors, *w*_FA_ has the most significant influence on *k*_c_, as shown in Table 7 and Figure 9. The *k*_c_ shows a significant decline with the increase of *w*_FA_. There are two mechanisms on the *k*_c_ reduction of FC by adding FA: (1) FA itself has lower *k*_c_ than cement; (2) the FA dosage can make the pores in FC more uniform, which is helpful for reducing the *k*_c_ [45]. It can be seen from Figure 8 that with the increase of *w*_FA_—although the connected holes increase and the pore diameter does not change too much—the pore wall becomes thinner, which makes the pores more compact and increases the porosity. Since the *k*_c_ of air is lower than that of other building materials, the higher the porosity, the lower the *k*_c_ is. Therefore, based on the above reasons, the *k*_c_ of HBFC decreases with the increase of *w*_FA_. With the increase in *w*_FOAM_, the pore size first decreased and then increased, but the number of close pores when the *w*_FOAM_ was 14.5 wt% was more than when the *w*_FOAM_ was 13.5 wt%. Previous researches have reported that porosity accounts for the main volume of FC and that the quality of pores affects the thermal insulation, with larger pore volume and finer pores contributing to better insulation [34,36]. Therefore, in general, with the increase of *w*_FOAM_, the *k*_c_ of HBFC gradually decreases with the increase of *W*/*B* and *w*_NSP_, and the *k*_c_ of HBFC arrives to an inflection point, as shown in Figure 7. When the *W*/*B* is 0.5 and *w*_NSP_ is 0.8%, a uniform and stable FC can be obtained, and the bubble can be distributed uniformly and retain stable to a maximum extent, rendering low *k*_c_.

### 4.6. Analysis of Variances Comprehensive Evaluation on the Optimal Ratio

In order to investigate which factor significantly affects the *ρ*_d_, *f*_cu_, and *k*_c_ values and select the optimal ratio of HBFC, the ANOVA were studied and the results shown in Table 8.

The number of levels (4) subtracted by 1 gave the degree of freedom (*DOF*). The sum of squares (*SS*_i_) is acquired by Equation (5) [25]:(5)$$\begin{array}{l} SS_{i}=4({\left(k_{1i}-k\right)}^{2}+{\left(k_{2i}-k\right)}^{2}+{\left(k_{3i}-k\right)}^{2}+{\left(k_{4i}-k\right)}^{2})\\ \left(i=A,B,C,D\right) \end{array}$$
where *SS*_i_ stands for the sum of squares and *k* stands for the 16 mean values. Sum of squares divided by the degree of freedom produced mean squares (*M*_i_).

As demonstrated in Table 8, the minimum mean square (*M*_i_^min^ = 243.0) is chosen as error, other *M*_i_ divided by *M*_i_^min^ gives the variance ratios (*VR*_i_) of *ρ*_d_. Here, because of 1.00 < 1.19 < 3.37 < 3.67 < *F*_0.1_(3,3) = 5.39, the effects of these four factors on *ρ*_d_ are not significant, and the order of importance is *C* > *B* > *A* > *D*.

As shown in Table 7 and Figure 5, the difference in *ρ*_d_ of HBFC under different factors and levels is little, but when factor *A* is at level 2, factor *B* is at level 1, factor *C* is at level 4, and factor *D* is at level 1, the minimum *ρ*_d_ can be obtained, which are 314.8 kg/m^3^, 308.0 kg/m^3^, 307.2 kg/m^3^, and 319.0 kg/m^3^, respectively.

Consequently, the optimum parameters for *ρ*_d_ are *w*_FA_ at 10 wt%, *W*/*B* at 0.4, *w*_FOAM_ at 16.5 wt%, and *w*_NSP_ at 0 wt%. As for *f*_cu_, the minimum mean square (*M*_i_^min^ = 0.011) is chosen as the error, other *M*_i_ divided by *M*_i_^min^ gives the *VR*_i_ of *f*_cu_. As 1.00 < 2.09 < 2.64 < 3.36 < *F*_0.1_(3,3) = 5.39, the effects of these four factors on *f*_cu_ are also not significant, and the order of importance is *D* > *A* > *C* > *B.* From Table 7 & Figure 5, the optimum parameters for *f*_cu_ can be obtained, which are *w*_FA_ at 0 wt%, *W*/*B* at 0.45, *w*_FOAM_ at 14.5 wt%, and *w*_NSP_ at 0.6 wt%. The minimum mean square (*M*_i_^min^ = 4.89 × 10^−5^) is chosen as the error, other *M*_i_ divided by *M*_i_^min^ gives the *VR*_i_ of *k*_c_. As 20.86 > *F*_0.05_(3,3) = 9.28 and 174.23 > *F*_0.01_(3,3) = 29.46, factor *A* is the most significant factor on *k*_c_, and factor *C* is more significant than factors *B* and *D*. Similarly, the optimum parameters for *f*_cu_ can be obtained, which are *w*_FA_ at 20 wt%, *W*/*B* at 0.5, *w*_FOAM_ at 16.5 wt%, and *w*_NSP_ at 0 wt%. The optimal solution based on each performance of HBFC is shown in Table 9.

It can be seen from the ANOVA in Table 8, the effect of *w*_FA_ and *w*_FOAM_ on the *k*_c_ of HBFC is particularly significant. The order of importance of the three performance indictors is *k*_c_ > *f*_cu_ > *ρ*_d_. A comprehensive evaluation will be used to seek the optimal proportion for simultaneously satisfying the properties of the three types.

Factor *A*: For factor *A*, the optimal levels to satisfy three performances are *A*_2_, *A*_1_, and *A*_4_, respectively, with no overlapping levels as shown in Table 9. The *k*_c_ is more important than the other two properties. It can be seen from Table 7, the *k*_c_ at *A*_4_ level is 0.11604 W/m·k, which is better than levels *A*_1_ and *A*_2_. FA is cheaper than cement and conforms to the concept of green environmental protection. Level *A*_4_ satisfies the requirements of *ρ*_d_ and *f*_cu_, which are 322.9 kg/m^3^ and 0.58 MPa, respectively, as shown in Table 7. After comprehensive consideration, level *A*_4_ (*w*_FA_ = 20 wt%) will be selected.

Factor *B*: The *k*_c_ of level *B*_3_ is 0.15581 W/m·k, which is better than levels *B*_1_ and *B*_2_. As shown in Table 7, the *f*_cu_ of levels *B*_2_ and *B*_3_ are very similar, which are higher than level *B*_1_. And the *ρ*_d_ meet the requirements. Thus, level *B*_3_ (*W*/*B* = 0.5) is the optimal level.

Factor *C*: It can be seen from Table 9, there are two performance indicators inclined to level *C*_4_. The *k*_c_ and *ρ*_d_ under level *C*_4_ condition are better than *C*_2_, and the *f*_cu_ also meets the requirements. Therefore, level *C*_4_ (*w*_FOAM_ = 16.5 wt%) will be selected as the optimal level after comprehensive assessment.

Factor *D*: As shown in Table 9, there are two performance indicators inclined to level *D*_1_. But the *f*_cu_ of level *D*_2_ is 0.74 MPa, which is higher than level *D*_1_ whose *f*_cu_ is 0.52 MPa, and the *k*_c_ of levels *D*_1_ and *D*_2_ is almost equal. The *ρ*_d_ under level *D*_3_ also meets the requirement. Hereby, level *D*_2_ (*w*_NSP_ = 0.6 wt%) is selected as the optimal level.

After comprehensive assessment, the optimal proportion of HBFC is ultimately chosen as *A*_4_*B*_3_*C*_4_*D*_2_. The verification test demonstrates that, the *k*_c_, *f*_cu_, and *ρ*_d_ are 0.09234 ± 0.00142 W/m·k, 0.58 ± 0.02 MPa, and 293.5 ± 4.9 kg/m^3^, respectively, which effectively meet the requirements of ultra-light thermal insulation panels.

### 4.7. Cost and Scalability of HBFC

The raw materials for preparation of HBFC are HBSC, grade-II FA, a composite foaming agent, water, and NSP, and their market prices are 111.7 $/ton, 14.9 $/ton, 1,489.4 $/ton, 0.5 $/ton, and 4,468.3 $/ton. The production of 1 m^3^ HBFC that meets the performance criteria requires 200 kg of HBSC cement, 50 kg of grade-II FA, 6.5 kg of composite foaming agent, 80 kg of water, and 1.5 kg of NSP, and the total cost is about 39.54 $/1m^3^. As far as we know, the key to make ultra-light FC is to achieve the matching of binder condensation and foam defoaming. If OPC and FA are used to make FC that meets the above requirements for *f*_cu_ and *ρ*_d_, it is necessary to add an appropriate amount of accelerator and foam stabilizer, with the common-used fractions as 0.3 wt% and 0.6 wt% of the binder. The price of the OPC, accelerator, and foam stabilizer are 74.5 $/ton, 29,788.5 $/ton, and 4,468.3 $/ton, respectively. The total cost of 1 m^3^ OPC-based FC that meets the properties is about 44.7 $/m^3^.

Therefore, both in terms of physical properties and cost, the HBFC has superior advantages and cost-effectiveness than of OPC-based FC, and the pre-foaming method in this study is also helpful to scalability preparation of HBFC.

## 5. Conclusions

The 1h bleeding volume and 1 h settling distance of the foam increase with the dilution ratio of the foam. The foamability gradually decreases and the foam density gradually increases with the increase of dilution ratio. When the dilution ratio is higher than 1:30, the foam properties can meet the requirements. When the dilution ratio is 1:10, the foam stability is best, with 1 h bleeding volume and 1 h settling distance of the foam as 40.6 ± 2.1 mL and 4.7 ± 0.3 mm, respectively.

The *ρ*_d_ shows no regular trend with the change of four factors. In the mixing and curing process, the less the foam is broken, the higher the porosity of HBFC, and the lower the *ρ*_d_ of HBFC. In 16 groups of HBFCs, the minimum *ρ*_d_ is 294.4 ± 5.6 kg/m^3^. In general, the higher the *ρ*_d_, the higher the *f*_cu_ of HBFC. When the pore size of HBFC is small, the pore wall is thick, the connected pores are few, and the pore wall structure is strong, the *f*_cu_ of HBFC is more favorable. The highest *f*_cu_ can reach 1.05 ± 0.12 MPa. With the increase of *w*_FA_ and *w*_FOAM_, the porosity of HBFC can be improved, thus significantly reducing the *k*_c_ of HBFC. The lowest *k*_c_ in 16 groups can reach 0.09385 ± 0.00117 W/m·k.

From ANOM and ANOVA, the order of the factors affecting the *ρ*_d_, *f*_cu_, and *k*_c_ can be ranked in order of importance as *C* > *B* > *A* > *D* (*W*/*B* > *w*_FOAM_ > *w*_FA_ > *w*_NSP_, *D* > *A* > *C* > *B*, and *A* > *C* > *D* > *B,* respectively. Optimal mix proportion of *ρ*_d_, *f*_cu_, and *k*_c_ of HBFC based on orthogonal experimental are *A*_2_*B*_1_*C*_4_*D*_1_, *A*_1_*B*_2_*C*_2_*D*_2_, and *A*_4_*B*_3_*C*_4_*D*_1_, respectively.

With the increase in *w*_FOAM_, the micropore size increases and the uniformity deteriorates. When the *w*_FOAM_ is 16.5 wt%, the largest pore diameter can reach 450 μm, which undoubtedly improves the porosity of HBFC and is beneficial to reduce the *ρ*_d_ and *k*_c_ of HBFC. With the increase of *w*_FA_, the number of connected pores increases and the pore structures become looser, which is bad for the *f*_cu_.

Through comprehensive assessment, the optimal mix ratio of HBFC is *A*_4_*B*_3_*C*_4_*D*_2_. The *k*_c_, *f*_cu_, and *ρ*_d_ are 0.09234 ± 0.00142 W/m·k, 0.58 ± 0.02 MPa, and 293.5 ± 4.9 kg/m^3^, respectively, and the corresponding cost of HBFC is about 39.5 $/m^3^.

Future studies will focus on surface water repellent treatment of HBFC, to achieve lower *k*_c_, and effectively meet the thermal insulation requirements of prefabricated ultra-light building panels.

## Figures and Tables

**Figure 1 materials-12-00984-f001:**
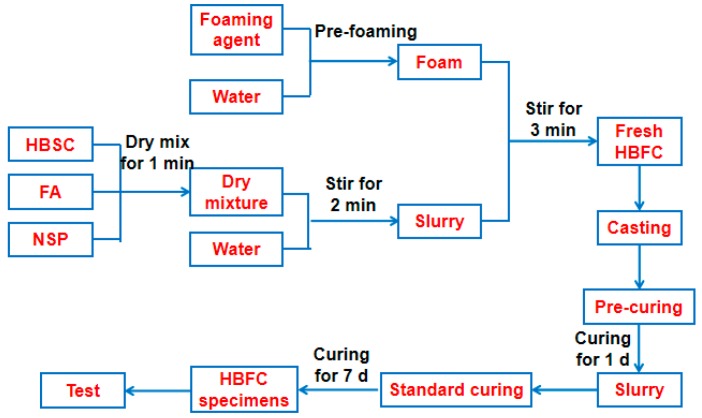
Schematic preparing process of HBFC.

**Figure 2 materials-12-00984-f002:**
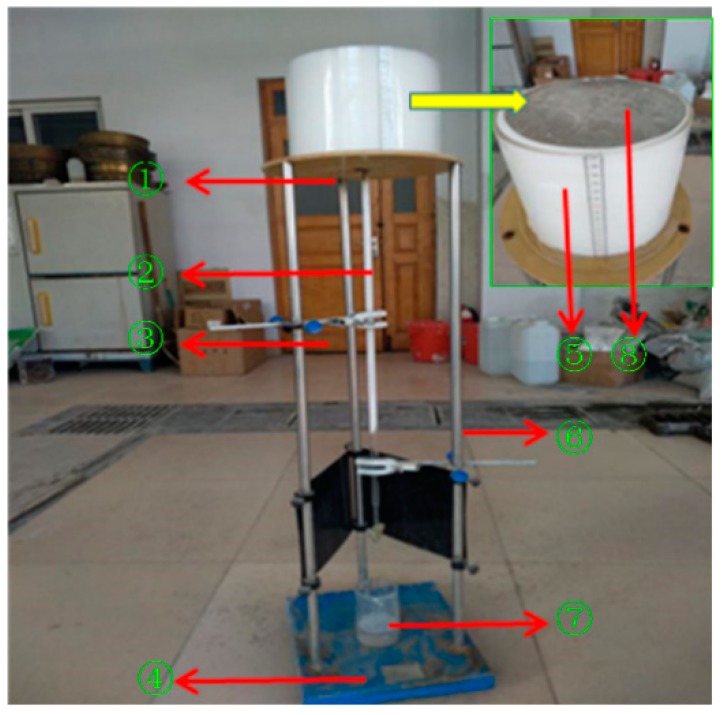
A foam tester for FC: **①**, Steel tray; **②**, long scale tube; **③**, catheter clip; **④**, base; **⑤**, capacity cylinder (enlarged green square); **⑥**, support bar; **⑦**, bleeding capacity bottle; **⑧****,** aluminum plate.

**Figure 3 materials-12-00984-f003:**
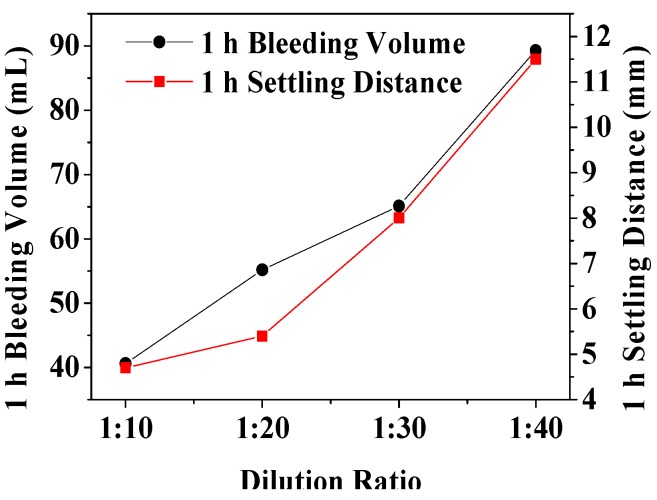
The relationship between 1 h bleeding volume, 1 h settling distance, and dilution ratio.

**Figure 4 materials-12-00984-f004:**
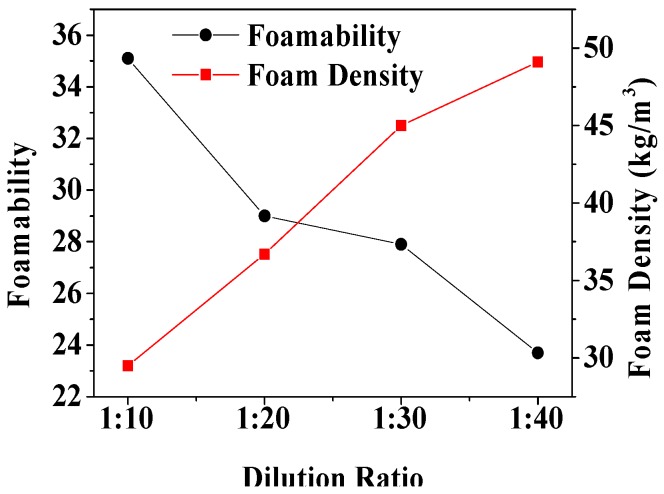
The relationship between foamability, foam density, and dilution ratio.

**Figure 5 materials-12-00984-f005:**
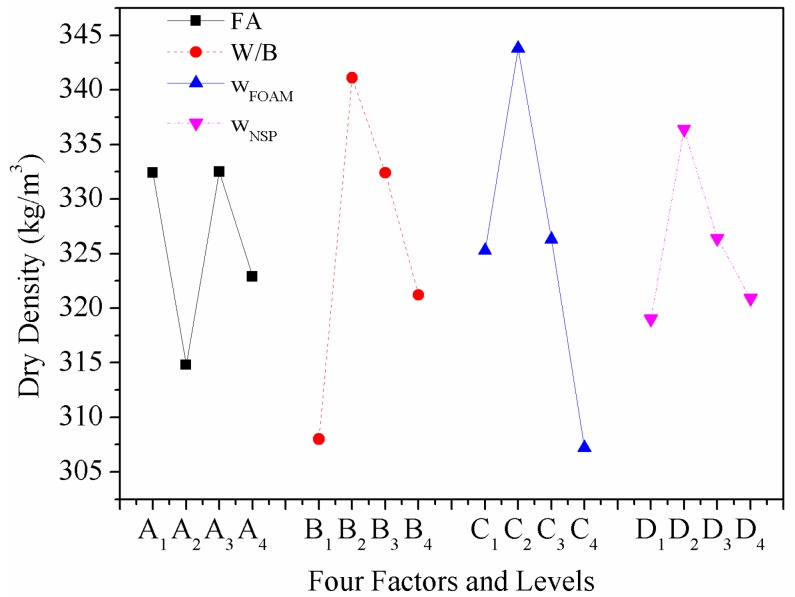
The influence of four factors on the dry density of HBFC.

**Figure 6 materials-12-00984-f006:**
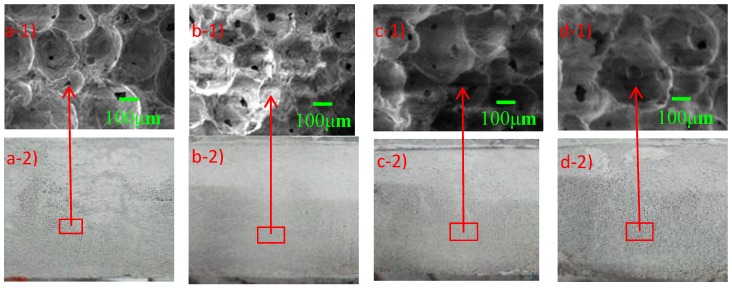
Pore size and naked surface images of HBFC under different *w*_FOAM_ observed by SEM and digital camera: (**a**) 13.5 wt%, (**b**) 14.5 wt%, (**c**) 15.5 wt%, and (**d**) 16.5 wt%.

**Figure 7 materials-12-00984-f007:**
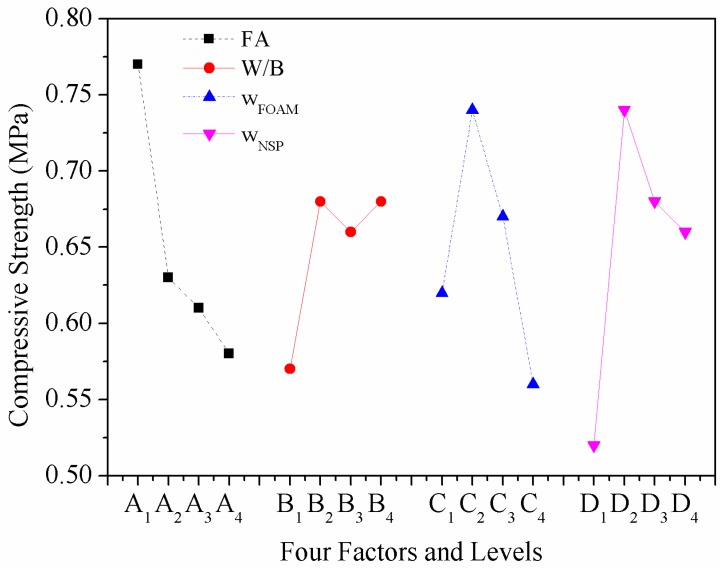
The influence of four factors on the compressive strength of HBFC.

**Figure 8 materials-12-00984-f008:**
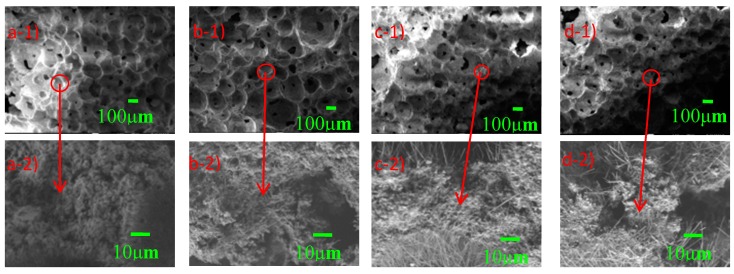
SEM morphology of pore wall structures of HBFC with different *w*_FA_: (**a**) 0 wt%, (**b**) 10 wt%, (**c**) 15 wt%, and (**d**) 20 wt%.

**Figure 9 materials-12-00984-f009:**
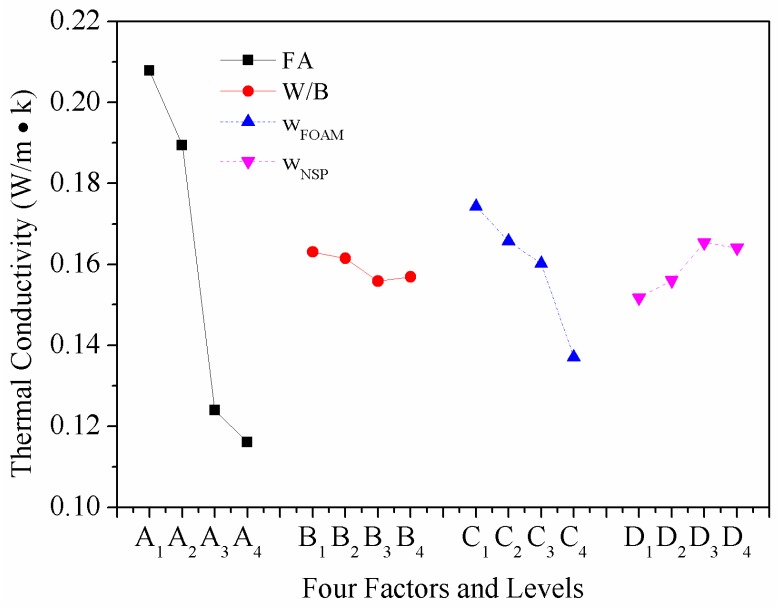
The influence of four factors on the thermal conductivity coefficient of HBFC.

**Table 1 materials-12-00984-t001:** Chemical compositions and mineral compositions of HBSC.

**Chemical Composition**	**CaO**	**SiO_2_**	**Al_2_O_3_**	**Fe_2_O_3_**	**MgO**	**SO_3_**	**TiO_2_**	**Sum**	**Loss**
	51.54	13.80	15.34	1.52	2.08	14.21	0.71	99.58	0.38
**Mineral Composition**	$C_{4}A_{3}\bar{S}$		**C_2_S**	**f-CaSO_4_**	**C_4_AF**		**f-CaO**	**CT**	
	29.35		38.06	13.64	5.08		1.84	1.11	

**Table 2 materials-12-00984-t002:** Chemical compositions of grade-II FA.

Ingredient	SiO_2_	Al_2_O_3_	Fe_2_O_3_	CaO	MgO	SO_3_	Na_2_O	K_2_O	Loss
FA	54.91	25.82	6.91	8.74	2.05	0.61	0.32	0.10	0.54

**Table 3 materials-12-00984-t003:** Factors and levels on preparation of HBFC.

	Factor	*A w*FA (wt%)	*B W*/*B*	*C w*FOAM (wt%)	*D w*NSP (wt%)
Level					
1		0	0.40	13.5	0
2		10	0.45	14.5	0.6
3		15	0.50	15.5	0.8
4		20	0.55	16.5	1.0

**Table 4 materials-12-00984-t004:** *L*_16_(4^4^) Orthogonal experimental array on mix proportion of HBFC.

	Factor	*A_**w*_FA_ (wt%)	*B_**W*/*B*	*C_**w*_FOAM_ (wt%)	*D_**w*_NSP_ (wt%)
Item No.					
*d* _1_		0 (1)	0.40 (1)	13.5 (1)	0% (1)
*d* _2_		0 (1)	0.45 (2)	14.5 (2)	0.6% (2)
*d* _3_		0 (1)	0.50 (3)	15.5 (3)	0.8% (3)
*d* _4_		0 (1)	0.55 (4)	16.5 (4)	1% (4)
*d* _5_		10 (2)	0.40 (1)	14.5 (2)	0.8% (3)
*d* _6_		10 (2)	0.45 (2)	13.5 (1)	1% (4)
*d* _7_		10 (2)	0.50 (3)	16.5 (4)	0% (1)
*d* _8_		10 (2)	0.55 (4)	15.5 (3)	0.6% (2)
*d* _9_		15 (3)	0.40 (1)	15.5 (3)	1% (4)
*d* _10_		15 (3)	0.45 (2)	16.5 (4)	0.8% (3)
*d* _11_		15 (3)	0.50 (3)	13.5 (1)	0.6% (2)
*d* _12_		15 (3)	0.55 (4)	14.5 (2)	0% (1)
*d* _13_		20 (4)	0.40 (1)	16.5 (4)	0.6% (2)
*d* _14_		20 (4)	0.45 (2)	15.5 (3)	0% (1)
*d* _15_		20 (4)	0.50 (3)	14.5 (2)	1% (4)
*d* _16_		20 (4)	0.55 (4)	13.5 (1)	0.8% (3)

**Table 5 materials-12-00984-t005:** Foam properties of composite foam agents under different dilution ratios.

Dilution Ratio	*B*_v_ (mL)	*S*_v_ (mm)	*M*	*ρ*_F_ (kg/m^3^)
1:10	40.6 ± 2.1	4.7 ± 0.3	35.1 ± 1.2	29.5 ± 0.9
1:20	55.2 ± 2.8	5.4 ± 0.3	29.0 ± 1.6	36.7 ± 1.1
1:30	65.1 ± 3.5	8.0 ± 0.4	27.9 ± 0.9	45.0 ± 1.4
1:40	89.3 ± 4.0	11.5 ± 0.6	23.7 ± 1.4	49.1 ± 1.4
JG/T 266-2011	<80	<10	> 20	-

**Table 6 materials-12-00984-t006:** Physical performances of HBFC based on the orthogonal test.

Item No.	*ρ*_d_ (kg/m^3^)	*f*_cu_ (MPa)	*k*_c_ (W/m·k)
*d* _1_	307.3 ± 3.5	0.49 ± 0.04	0.22368 ± 0.00121
*d* _2_	387.0 ± 4.9	1.05 ± 0.12	0.21544 ± 0.00098
*d* _3_	341.0 ± 5.0	0.82 ± 0.09	0.21098 ± 0.00144
*d* _4_	294.4 ± 5.6	0.73 ± 0.09	0.18131 ± 0.00109
*d* _5_	305.7 ± 4.8	0.67 ± 0.05	0.19903 ± 0.00209
*d* _6_	325.5 ± 7.0	0.63 ± 0.02	0.21098 ± 0.00087
*d* _7_	306.8 ± 3.3	0.47 ± 0.02	0.15835 ± 0.00174
*d* _8_	321.2 ± 4.0	0.73 ± 0.04	0.18920 ± 0.00113
*d* _9_	321.0 ± 2.9	0.62 ± 0.06	0.13561 ± 0.00103
*d* _10_	329.7 ± 3.9	0.54 ± 0.01	0.11446 ± 0.00144
*d* _11_	339.2 ± 6.5	0.69 ± 0.04	0.12552 ± 0.00069
*d* _12_	339.9 ± 7.0	0.59 ± 0.02	0.12001 ± 0.00144
*d* _13_	298.0 ± 4.1	0.50 ± 0.02	0.09385 ± 0.00117
*d* _14_	322.0 ± 3.3	0.51 ± 0.02	0.10492 ± 0.00069
*d* _15_	342.5 ± 4.4	0.65 ± 0.06	0.12837 ± 0.00095
*d* _16_	329.1 ± 6.6	0.67 ± 0.04	0.13703 ± 0.00126

**Table 7 materials-12-00984-t007:** ANOM of the orthogonal test and optimization of the mix proportion of HBFC.

Index	Factor	*A*	*B*	*C*	*D*	
*ρ*_d_ (kg/m^3^)	*k* _1_	332.4	308.0	325.3	319.0	Optimal mix proportion: *A*_2_*B*_1_*C*_4_*D*_1_
	*k* _2_	314.8	341.1	343.8	336.4	
	*k* _3_	332.5	332.4	326.3	326.4	
	*k* _4_	322.9	321.2	307.2	320.9	
	Range	17.65	33.05	36.55	17.35	
	Optimal level	2	1	4	1	
	Factor order	*C* > *B* > *A* > *D*				
*f*_cu_ (MPa)	*k* _1_	0.77	0.57	0.62	0.52	Optimal mix proportion: *A*_1_*B*_2_*C*_2_*D*_2_
	*k* _2_	0.63	0.68	0.74	0.74	
	*k* _3_	0.61	0.66	0.67	0.68	
	*k* _4_	0.58	0.68	0.56	0.66	
	Range	0.19	0.11	0.18	0.23	
	Optimal level	1	2	2	2	
	Factor order	*D* > *A* > *C* > *B*				
*k*_c_ (W/m·k)	*k* _1_	0.20785	0.16304	0.17430	0.15174	Optimal mix proportion: *A*_4_*B*_3_*C*_4_*D*_1_
	*k* _2_	0.18939	0.16145	0.16571	0.15600	
	*k* _3_	0.1239	0.15581	0.16018	0.16537	
	*k* _4_	0.11604	0.15689	0.13699	0.16407	
	Range	0.09181	0.00723	0.03731	0.01364	
	Optimal level	4	3	4	1	
	Factor order	*A* > *C* > *D* > *B*				

**Table 8 materials-12-00984-t008:** ANOVA results according to *ρ*_d_, *f*_cu_, and *k*_c_ shown in Table 4 and Table 6.

Index	Factor	*A*	*B*	*C*	*D*	Error
*ρ*_d_ (kg/m^3^)	*DOF*	3	3	3	3	3
	*SS* _i_	869.7	2456.6	2674.2	729.1	729.1
	*M* _i_	289.9	818.9	891.4	243.0	243.0
	*VR* _i_	1.19	3.37	3.67	1.00	1.00
*f*_cu_ (MPa)	*DOF*	3	3	3	3	3
	*SS* _i_	0.087	0.034	0.070	0.110	0.034
	*M* _i_	0.029	0.011	0.023	0.037	0.011
	*VR* _i_	2.64	1.00	2.09	3.36	1.00
*k*_c_ (W/m·k)	*DOF*	3	3	3	3	3
	*SS* _i_	2.55 × 10^−2^	1.47 × 10^−4^	3.06 × 10^−3^	5.11 × 10^−4^	1.47 × 10^−4^
	*M* _i_	8.52 × 10^−3^	4.89 × 10^−5^	1.02 × 10^−3^	1.70 × 10^−4^	4.89 × 10^−5^
	*VR* _i_	174.23 **	1.00	20.86 *	3.48	1.00

Noting that, *F*_0.1_(3,3) = 5.39, *F*_0.01_(3,3) = 29.46; *, ** represent significant, more significant factor, respectively.

**Table 9 materials-12-00984-t009:** Optimal solutions of various factors based on the performance of HBFC.

Performance	Factors and Level			
*ρ* _d_	*A* _2_	*B* _1_	*C* _4_	*D* _1_
*f* _cu_	*A* _1_	*B* _2_	*C* _2_	*D* _2_
*k* _c_	*A* _4_	*B* _3_	*C* _4_	*D* _1_

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
