# Peer review of "Preparation and Physical Properties of High-Belite Sulphoaluminate Cement-Based Foam Concrete Using an Orthogonal Test"

_materials, 2019, doi:10.3390/ma12060984_

Round 1
Reviewer 1 Report
The paper concerns the investigations about foam concrete and its properties. The experiment is well designed. In addition to purely material research (e.g., compressive strength, density, thermal conductivity), the research is supported by statistical analysis (e.g., ANOM method) which makes the paper more valuable. The paper is properly organized and presents high scientific and practical value. In spite of all, the paper requires revision before acceptance for publication. Detailed comments are listed below:
1. The paper should be checked by native English speaker in order to improve the style; there is also few syntax and grammar errors.
2. Line 53-56 – the authors wrote some info about the use of CNTs to foam concrete; the authors may add that sometimes the use of CNTs for concrete together with surfactant (e.g. SDS), which is intended to disperse CNTs in the volume of a cement matrix, may also cause its foaming, an example reference in this subject:
Szeląg M.: Mechano-physical properties and microstructure of carbon nanotube reinforced cement paste after thermal load. Nanomaterials, vol. 7(9), 2017, 267
Liu, Q.; Sun,W.; Sun, B.; Sun, X.; Jia, L.; Ma, Z. Preparation of carbon nanotubes solution and its effects on mechanical properties of cement mortar. J. Southeast Univ. (Nat. Sci. Ed.) 2014, 44, 662–667
3. Tab. 1 and 2 - how was the chemical composition tested? Please specify the method or add info if the data was received from the manufacturer.
4. Caption of Fig. 4 – maybe not “intelligent”, it would be more appropriate to use the “automatic”.
5. Line 130-132 - please pay attention that the formatting should be in accordance with the requirements of the journal.
6. Thermal conductivity section - for the future I recommend testing the thermal conductivity coefficient in the lower set of temperatures; testing at 40/20 °C gives an average system temperature of 30 °C; generally, the thermal conductivity coefficient increases to a small extent as the material temperature rises, which can overstate the result in real conditions where high thermal insulation is required, i.e. at temperatures of less than 10 °C.
7. Tab. 5 and 6 - how many samples were tested in each test? Please provide the basic data on the distribution of results, e.g., random coefficient of variation or standard deviation; it is necessary to determine the repeatability and reliability of the results obtained.
Author Response
Dear the Reviewer,
We would like to express our sincere appreciations for your valuable comments and suggestions to improve the quality of this paper. The responses are mentioned as below, and details changes as per new manuscript are correspondingly tracked in tandem with substantial improvements on English language and styles.
With best regards,
Jianlin Luo
On behalf of the author team

Reviewer 2 Report
The authors should consider some changes before acceptance:
All currencies regarding the cost analysis should be converted to Euro or US dollar.
Please provide much more important results in abstract compared to report the mix composition of the best mixture.
Acheivingto uniform porosity is acheived mostly in the foamed concrete using pre-made foaming approach. Replace this result in abstract with more important one.
Remove figures 1 and 4. they are really low quality figures.
Improve the quality of figures 2 and 3.
Author Response
Dear the Reviewer,
We would like to express our sincere appreciations for your valuable comments and suggestions to improve the quality of this paper. The responses are mentioned as below, and the corresponding revisions are highlighted and tracked in the revised edition along with substantial improvements on English language and styles.
With best wishes,
Jianlin Luo
On behalf of the author team

Round 2
Reviewer 1 Report
All suggestions have been included and the paper is in much better quality. I accept the paper for publication.
Reviewer 2 Report
Now, paper can be accepted.